# Community Strategy for Hepatitis B, C, and D Screening and Linkage to Care in Mongolians Living in Spain

**DOI:** 10.3390/v15071506

**Published:** 2023-07-05

**Authors:** Adriana Palom, Edurne Almandoz, Antonio Madejón, Ariadna Rando-Segura, Ylenia Pérez-Castaño, Judit Vico, Sara Gándara, Naranbaatar Battulga, Jordi Gómez-I-Prat, Mar Riveiro-Barciela, Juan Arenas Ruiz-Tapiador, Javier García-Samaniego, Maria Buti

**Affiliations:** 1Departamento de Hepatología, Hospital Universitari Vall d’Hebron, 08035 Barcelona, Spain; adrianapalom@gmail.com (A.P.); judviro1630@gmail.com (J.V.); sara.gandara.diez@gmail.com (S.G.); baadaisuld@gmail.com (N.B.); mar.riveiro@gmail.com (M.R.-B.); 2Centro de Investigación Biomédica en Red en Enfermedades Hepáticas y Digestivas, 28029 Madrid, Spain; amadejon@yahoo.com (A.M.); ariadna.rando@vallhebron.cat (A.R.-S.); juan.arenasruiz@gmail.com (J.A.R.-T.); javiersamaniego@telefonica.net (J.G.-S.); 3Departamento de Enfermedades Gastrointestinales, Hospital Universitario Donostia, Instituto de Investigación Biodonostia, 20014 San Sebastian, Spain; edurne.almandoz@biodonostia.org; 4Departamento de Hepatología, Hospital Universitario La Paz, IdiPAZ, 28046 Madrid, Spain; 5Departamento de Microbiología, Laboratorios Clínicos, Hospital Universitario Vall d’Hebron, 08035 Barcelona, Spain; 6Departamento de Digestivo, Hospital Bidasoa de Hondarribia, Instituto de Investigación Biodonostia, 20280 San Sebastian, Spain; ylenia_perez@hotmail.com; 7Equipo de Salud Pública y Comunitaria (ESPIC), Unidad de Salud Internacional Drassanes-Hospital Vall d’Hebron, 08001 Barcelona, Spain; jordi.gomez@vallhebron.cat; 8Policlínica Gipuzkoa-Quiron Salud, 20012 San Sebastian, Spain

**Keywords:** community program, viral hepatitis screening, viral hepatitis awareness, Mongolian community

## Abstract

Mongolia has one of the highest viral hepatitis infection (B, C, and D) rates in the world. The aims of this study were to increase awareness of this disease and promote viral hepatitis screening in the Mongolian community living in Spain. Through a native community worker, Mongolian adults were invited to a community program consisting of an educational activity, an epidemiological questionnaire, and rapid point-of-care testing for hepatitis B and C. In those testing positive, blood extraction was performed to determine serological and virological parameters. In total, 280 Mongolians were invited to the program and 222 (79%) attended the event: 139 were women (63%), mean age was 42 years, and 78 (35%) had viral hepatitis risk factors. Testing found 13 (5.8%) anti-HCV-positive individuals, 1 with detectable HCV RNA (0.5%), 8 HBsAg-positive (3.6%), and 7 with detectable HBV DNA (3.1%). One additional individual had HBV/HCV co-infection with detectable HBV DNA and HCV RNA. Two subjects had hepatitis B/D co-infection (0.9%). The knowledge questionnaire showed a 1.64/8-point (20.5%) increase in correct answers after the educational activity. In summary, a viral hepatitis community program was feasible and widely accepted. It increased awareness of this condition in the Mongolian community in Spain and led to linkage to care in 22 participants, 50% of whom were unaware of their infection.

## 1. Introduction

In 2021, the World Health Organization (WHO) reported more than 800,000 cases of liver cancer worldwide due to viral hepatitis, with the main causes being chronic hepatitis B virus (HBV) and hepatitis C virus (HCV) infection [1]. Chronic hepatitis D virus (HDV) infection, which occurs in the presence of HBV, causes the most severe form of viral liver disease, with faster and more frequent progression to liver cirrhosis [2]. Mongolia has the highest incidence of liver cancer in the world, mainly due to chronic HBV and HCV infection. It is the first lower/middle-income country in Asia or the Pacific region to commit to hepatitis elimination, in keeping with the WHO global hepatitis strategy [3,4].

In the 2015 WHO report of viral hepatitis in Mongolia, the reported prevalence of hepatitis C antibody (anti-HCV) was 11.1–15.6%, hepatitis B surface antigen (HbsAg) was 10.6%, and hepatitis D antibody (anti-HDV) 41–67% among those HbsAg-positive [5].

In a more recent nationwide serosurvey performed in Mongolia in 2022, which included 10,040 participants from 10 to 64 years old, the prevalence was anti-HCV 8.9%, HBsAg 4.3%, and anti-HDV 4.8%. More than 60% of affected participants were unaware of having HBV or HCV infection, and 99% were unaware of their HDV-positive status [6]. In 2021, retrospective point-of-care testing in 251 HBsAg-positive serum samples from adults in Inner Mongolia detected anti-HDV in 13.4%, with detectable HDV RNA in 97% [7].

Studies investigating viral hepatitis in Mongolians who have migrated to another country have reported similar rates of these infections than in the native population. A recent serosurvey conducted in the Mongolian population living in Southern California (USA) reported an anti-HCV prevalence of 9.1%, HBsAg 9.7%, and 41.2% (21/51) of positive anti-HDV among HBsAg-positive participants. HDV RNA was detectable in 81% (17/21), and this group had a statistically greater degree of liver fibrosis than those with undetectable HDV RNA [8]. In another study, performed in Mongolians living in Washington DC, the prevalence of HBsAg was 6.2%, and anti-HCV was 9.9% [9].

Around 107,140 Mongolian nationals were reported to be living abroad in the Mongolian census of 2010. An estimated 825 Mongolian nationals are currently living in Spain, with 698 (85%) of them located in three main areas: Barcelona, San Sebastian, and Madrid [10]. Vaccination against HBV in Mongolia was incorporated into the Expanded Program on Immunization in 1991 [11]. This effort resulted in a significant decrease in the prevalence of HBV in children [12], but unvaccinated adults are still at risk of acquiring HBV infection and developing hepatocellular carcinoma (HCC) [13].

Taking integrative programs out of the medical setting and into the community can generate interest and improve awareness of viral hepatitis infection and its consequences. Educational tools such as informative videos and knowledge questionnaires have proven to be beneficial in community screening efforts [14]. Commercially available, fast, and easy-to-use diagnostic tests for HBV and HCV using blood drop samples are also very effective in this setting. This point-of-care testing method has been widely used, especially in low/middle-income communities and vulnerable populations [15,16].

The aim of this study was to develop a community program to increase awareness of viral hepatitis in the Mongolian population living in Spain, and to use point-of-care diagnostic tools to identify, characterize, and link to care those who need it. 

## 2. Materials and Methods

### 2.1. Study Design

Prospective community program consisting of an educational activity, demographic questionnaire, and rapid hepatitis B and C testing performed from April to November in 2022. The professionals involved included experienced community workers, nurses, and physicians. 

### 2.2. Participants

All individuals from Mongolia older than 18 years living in Spain and able to read and sign an informed consent (in Spanish or Mongolian) were prospectively invited to participate in this study. Those unable to speak Spanish or Mongolian were excluded.

### 2.3. Methods

Mongolian adults residing in Spain were contacted by our Mongolian community worker who advertised the activity through social media and sent an invitation to participate. The steps comprising the community-based program are represented in Figure 1. Participants signed an informed consent form and answered a baseline questionnaire that included socio–demographic data, risk factors, previous access to healthcare, previous viral hepatitis testing (self-reported infection status or previous diagnosis), and willingness to attend a medical visit if the viral hepatitis serology was positive. 

The proposed educational session was based on a modification of HEPARJOC [17], an informative tool designed to boost awareness of hepatitis B and improve accessibility to a clinical diagnosis in immigrant communities. HEPARJOC consists of five educational games, in which participants are guided by a facilitator, and a video presentation explaining the nature of viral hepatitis, the associated risk factors, and the importance of testing [14]. A subgroup of participants answered an eight-question survey translated into the Mongol language before and after the video to determine what they had learned about viral hepatitis (Appendix A). The data obtained in the two questionnaires were analyzed and compared. 

Participants underwent rapid HBsAg (Determine HBsAg2, Abbott, Lake Forest, IL, USA; limit of detection (LOD) 0.1 IU/mL) and anti-HCV (Anti-HCV Test WB/S/P, Türklab, Smyrna, Türkiye; sensitivity/specificity 100%) testing. A few drops of whole blood were collected onto special filter paper using a finger-stick technique. Reagents were then added, and 20 min later the results were read and recorded in a database. In those testing positive, a plasma separation card (PSC) test (Cobas) plasma separation card, Roche, Basel, Switzerland) was performed to analyze HBV DNA, HCV RNA, and anti-HDV. This is an easy-to-use sample collection method, currently approved for determining plasma HIV viral load. This unique technology separates plasma from whole blood obtained by finger-stick and creates a dry plasma spot for analysis. In parallel, blood was extracted to determine biochemical (ALT, AST, GGT), serological (HBsAg, anti-HCV, anti-HDV), and virological parameters (HBV DNA, LOD 2000 IU/mL; HCV RNA, LOD 1000 IU/mL) when applicable. After sample collection, PSCs and conventional blood samples were transported to our reference laboratory for analysis (Microbiology Department, Hospital Universitari Vall d’Hebron, Barcelona, Spain). All participants testing positive were linked to care on the same day as screening.

### 2.4. Ethical Considerations

This study was approved by the Research Ethics Committee of Vall d’Hebron Hospital and was conducted in compliance with the principles of the Declaration of Helsinki, Good Clinical Practice guidelines and local regulatory requirements. Informed consent forms were provided to all included subjects, and all data were anonymized.

### 2.5. Statistical Analysis

All statistical analyses were performed using IBM SPSS, version 26.0 (SPSS Inc., Armonk, NY, USA). Quantitative variables were expressed as median and interquartile range (IQR) and analyzed with the Mann–Whitney U test. Categorical variables were expressed as frequency and percentage and compared using the chi-squared or Fisher’s exact test, when frequencies were less than 5%. The results were considered statistically significant when the *p*-value was lower than 0.05. 

## 3. Results

Among 698 Mongolians residing in Barcelona, San Sebastian, and Madrid, 280 individuals were invited to the community program, and 222 (79%) agreed to participate and attended the event. All attendees performed the audiovisual activity, completed the demographic questionnaire, and underwent rapid testing. 

Baseline characteristics of the participants according to the questionnaire results are shown in Table 1. Most were women, originally from Ulaanbaatar city, employed full-time, and in possession of a secondary level of education. Risk factors were found in 35% of attendees, the most common being tattoos (25%), having received a blood transfusion (10%), and vertical transmission (8%). In addition, 22% of participants had been vaccinated against HBV. 

### Viral Hepatitis Results and Risk Factors

Among all participants, 13 (5.8%) were anti-HCV positive and 1 (0.5%) had detectable HCV RNA; 8 (3.6%) were HBsAg positive and 7 (3.1%) had detectable HBV DNA. Two of the total nine HBsAg-positive participants tested positive for anti-HDV (22%); HDV RNA was undetectable in both cases. One individual had HBV/HCV co-infection with detectable HBV DNA and HCV RNA (female, 34 years old, no risk factors, and unknown HBV vaccination status).

Individuals testing positive for HBV had a family member with known HBV more often than the total included. Participants who tested positive for HCV had a larger number of risk factors than HBV-positive individuals or the total tested. Overall, subjects with HBsAg or anti-HCV were older than those negative.

Among the 22 patients diagnosed, 11 (50%) were unaware of their diagnosis. All of them are currently linked to care. 

Results from PSC tests and conventional blood testing were concordant in all cases. HBV DNA determination from only one PSC was invalid due to insufficient sample. However, PSC quantification was slightly higher than blood test findings for HBV DNA levels (+0.72 log) and slightly lower for HCV RNA levels (−1.13 log).

The eight-question viral hepatitis knowledge questionnaire was performed in 89 randomly selected participants. Each question accounted for one point. The average number of correct responses was 4.7 out of 8 points before the activity, and 6.3 out of 8 afterward. There was a mean 1.6-point increase in correct answers after seeing the educational video. Only 9 (10.1%) participants answered all questions correctly before the activity, whereas 19 (21.3%) responded correctly to all questions after.

## 4. Discussion

This study, targeting the Mongolian community in Spain, illustrates how a population at high risk of viral hepatitis can be approached by the health system to provide care for their disease. The first aspect to highlight is the wide acceptance of the program, as almost 80% of individuals contacted accepted to participate. Thus, the program was feasible and well received, even in a community with certain language barriers, likely owing to the participation of a community worker who spoke the same language.

The program enabled identification of 13 (5.8%) anti-HCV-positive individuals, 1 with detectable HCV RNA (0.5%), 8 HBsAg-positive individuals (3.6%), and 7 with detectable HBV DNA (3.2%). Two participants had hepatitis B/D co-infection (0.9%), both with undetectable HDV RNA. One individual had HBV/HCV co-infection with detectable HBV DNA and HCV RNA.

Overall, the prevalence of viral hepatitis found in this study is lower than the rates reported in Mongolia. There are several potential reasons for this difference. First, Mongolian people who migrate to other countries are likely to be younger and healthier than those who stay. Second, hepatitis B vaccination to immunize children in Mongolia launched in 1991, and the use of disposable syringes since the 1990s have been huge breakthroughs in fighting transmission. In addition, in 2016, the Mongolian government implemented the National Viral Hepatitis Program, a comprehensive initiative involving all aspects from prevention to care and disease control, to meet a goal for reducing morbidity and mortality due to HBV, HCV, and HDV [18].

However, other studies examining viral hepatitis in Mongolians who have relocated outside of their native country, have revealed more comparable infection rates to those observed in the local population. A recent serosurvey conducted among Mongolian individuals residing in Southern California (USA) found a 9.1% prevalence of anti-HCV, 9.7% HBsAg, and 41.2% (21 out of 51) tested positive for anti-HDV among those HBsAg-positive [8]. Another study conducted among Mongolians living in Washington DC reported a prevalence of 6.2% for HBsAg and 9.9% for anti-HCV [9]. Viral hepatitis rates outside of Mongolia might be lower than those previously reported in other community programs, so more studies would be needed in order to stablish the cause of these differences.

As would be expected, the prevalence of viral hepatitis infection observed in this Mongolian screening program was higher than the values reported in the native Spanish population (0.85% for anti-HCV, 0.22% for HBsAg, and 7.7% for anti-HDV (among HBsAg-positive individuals)) [19,20]. These results further support the value of screening in migrant populations from countries with a high prevalence of this disease.

The present study was not intended to be a test-and-treat program. The educational aspect of the program was found to be effective. All attendees accepted participating in the HEPARJOC audiovisual activity with the assistance of the community worker, who additionally adopted a peer educator function. After the video, there was a 1.6-point improvement in correct answers on the knowledge questionnaire, an increase in line with that seen in the pilot study that validated HEPARJOC for hepatitis B [14]. In accordance with results from a recent study in a migrant population from Pakistan [21], HEPARJOC exerted a significant influence on our attendees, as it facilitated comprehension of the screening process and inculcated the importance of hepatitis testing.

As viral hepatitis infection is often asymptomatic and hepatitis screening is usually performed in a hospital setting, many cases can remain undiagnosed. It is particularly useful to offer educational activities on this disease in the community to at-risk groups with language and cultural differences to improve diagnostic rates. Bringing programs out of the medical setting and into the community can raise awareness of viral hepatitis infection and its consequences, and lead to better management of the disease. 

The major limitation of this study was that the program was structured with a limited number of sessions, which prevented some participants from attending due to geographical reasons, as they lived far from the session location.

In summary, the community program was feasible, widely accepted, and allowed linkage to care for 22 individuals with hepatitis infection, 50% of whom were unaware of the disease. Among the total of participants, the prevalence of hepatitis B was higher than active hepatitis C. Through the epidemiological questionnaire, it was assessed that a large number of adults were unaware if they had been vaccinated against HBV. In the future, hepatitis B vaccination should be included in these community programs.

## Figures and Tables

**Figure 1 viruses-15-01506-f001:**
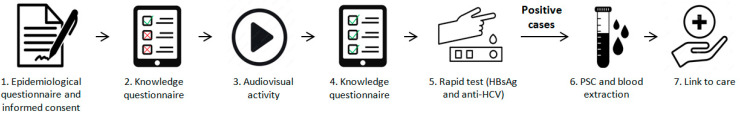
Steps comprising the community program.

**Table 1 viruses-15-01506-t001:** Baseline characteristics of the study participants.

	Total (N = 222)	Negative Serology (N = 200)	Positive Serology (N = 22) *		*p* Value **
			HBsAg (N = 8)	Anti-HCV (N = 13)	
Female	139 (63%)	125 (63%)	4 (50%)	9 (69%)	0.443
Age	42.0 (33.8–49.2)	42.0 (33.8–49.2)	44.6 (37.9–52.6)	51.7 (47.8–60.7)	0.011
Region origin					0.593
City	160 (72%)	145 (73%)	5 (62%)	9 (69%)	
Rural	53 (24%)	47 (24%)	3 (38%)	3 (23%)	
No data	9 (4%)	8 (4%)	0 (0%)	1 (8%)	
Years living in Spain	10 (5–13)	10 (5–13)	7 (3–12)	9 (7–16)	0.294
Employment status					0.175
Full-time	97 (44%)	88 (44%)	4 (50%)	4 (31%)	
Part-time	55 (25%)	50 (25%)	2 (25%)	3 (23%)	
Unemployed	55 (25%)	49 (25%)	2 (25%)	4 (31%)	
Retired	5 (2%)	3 (2%)	0 (0%)	2 (15%)	
No data	10 (4%)	10 (5%)	0 (0%)	0 (0%)	
Educational level					0.41
Primary	6 (3%)	6 (3%)	0 (0%)	0 (0%)	
Secondary	119 (53%)	105 (53%)	5 (62%)	8 (62%)	
Superior	86 (39%)	78 (39%)	3 (38%)	5 (38%)	
No data	11 (5%)	11 (6%)	0 (0%)	0 (0%)	
BMI	25.0 (22.6–27.8)	25.0 (22.6–27.8)	25.4 (23.1–29.3)	26.2 (25.3–29.9)	0.424
Known HBV infection	6 (3%)	0 (0%)	6 (75%)	0 (0%)	<0.001
Known HCV infection	5 (3%)	0 (0%)	0 (0%)	5 (38%)	<0.001
Diabetes	8 (4%)	8 (4%)	0 (0%)	0 (0%)	0.437
Hypertension	31 (14%)	29 (15%)	1 (12%)	1 (8%)	0.42
Family member with known HBV infection	22 (10%)	18 (9%)	2 (25%)	2 (15%)	0.148
Risk factors	78 (35%)	67 (34%)	3 (38%)	7 (54%)	0.175
Blood transfusion	22 (10%)	17 (9%)	1 (12%)	3 (23%)	
Tattoos	55 (25%)	47 (24%)	3 (38%)	4 (31%)	
PWID	2 (1%)	0 (0%)	0 (0%)	2 (15%)	
MSM	1 (0.5%)	1 (0.5%)	0 (0%)	1 (8%)	
Vertical transmission	18 (8%)	14 (7%)	1 (12%)	2 (15%)	
HBV-vaccinated					0.244
Vaccinated	49 (22%)	46 (23%)	2 (25%)	1 (8%)	
Unknown	173 (78%)	154 (77%)	6 (75%)	12 (92%)	

BMI, body mass index; MSM, men who have sex with men; PWID, people who inject drugs. * One subject was excluded from Table 1 for HBV/HCV co-infection. ** *p* values comprise the comparison between subjects with positive (HBsAg+ or anti-HCV+) and negative serology.

## Data Availability

The data presented in this study are available upon reasonable request to the corresponding author.

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
