# Peer review of "Community Strategy for Hepatitis B, C, and D Screening and Linkage to Care in Mongolians Living in Spain"

_viruses, 2023, doi:10.3390/v15071506_

Round 1
Reviewer 1 Report
Comments and Suggestions for Authors
This short study describes the status of hepatitis B, C, and D viruses infections in the Mongolian community living in Spain. Among 222 tested participants, the anti-HCV positivity was 5.8% (13/222) with 1 detectable HCV RNA; the HBsAg positivity was 3.6% (8/222) with 7 detectable HBV DNA; one individual had HBV/HCV co-infection with detectable HBV DNA and HCV RNA; and 2 of 9 HBsAg-positive patients were positive for anti-HDV but none detectable HDV RNA. Moreover, the educational activity increased by 1.6/8 points in correct answers to the questionnaire. Overall, the study is important to know the infection rates of viral hepatitis of Mongolians in Spain. However, the study is relatively simple with limited information. I have the following comments for the authors consideration.
1. The authors mentioned multiple times that Mongolia has high viral hepatitis rates in the world; however, the reviewer did not find any specific numbers of these high infection rates. Please specify in the maintext.
2. Is there any data regarding the viral hepatitis rates of Mongolians in other countries or areas? It would be beneficial to mention it in the Discussion section.
3. If I understand correctly, the 222 Mongolian participants were recruited from Barcelona, San Sebastian, and Madrid in Spain; what are the specific numbers of participants from these three cities? Curiously, could there be any difference in infection rates of viral hepatitis between cities?
4. The translated questionnaire is required to be presented as supplementary material for this study.
5. Is it possible to conduct any statistical analyses in Table 1? Which would enrich the results section.
6. Since this study focuses on viral hepatitis, why did the authors not include hepatitis A and E viruses?
Comments on the Quality of English LanguageMinor editing of English language required.
Author Response
Dear reviewer,
Thank you for your thoughtful response. We are pleased to know that our brief report is being considered for publication in your journal. We have carefully reviewed our manuscript according to your comments, which we believe have contributed to enriching the paper. Below you will find our point-by-point response to your suggestions, as well as the corresponding changes in the full text.
Reviewer #1: This short study describes the status of hepatitis B, C, and D viruses infections in the Mongolian community living in Spain. Among 222 tested participants, the anti-HCV positivity was 5.8% (13/222) with 1 detectable HCV RNA; the HBsAg positivity was 3.6% (8/222) with 7 detectable HBV DNA; one individual had HBV/HCV co-infection with detectable HBV DNA and HCV RNA; and 2 of 9 HBsAg-positive patients were positive for anti-HDV but none detectable HDV RNA. Moreover, the educational activity increased by 1.6/8 points in correct answers to the questionnaire. Overall, the study is important to know the infection rates of viral hepatitis of Mongolians in Spain. However, the study is relatively simple with limited information. I have the following comments for the authors consideration.
- The authors mentioned multiple times that Mongolia has high viral hepatitis rates in the world; however, the reviewer did not find any specific numbers of these high infection rates. Please specify in the maintext.
Thank you for your comment. The last 2015 WHO report for viral hepatitis in Mongolia has been added (page 2, paragraph 2). There are several serosurveys performed in Mongolia, we referenced the most recent nationwide data in the introduction (page 2, paragraph 3).
- Is there any data regarding the viral hepatitis rates of Mongolians in other countries or areas? It would be beneficial to mention it in the Discussion section.
In our knowledge, two US studies performed in Mongolians from South California and Washington DC assessed viral hepatitis infection rates in Mongolians, both papers being referenced in the introduction. We added a paragraph in the Discussion section to contrast our results with those mentioned (page 6, third paragraph).
- If I understand correctly, the 222 Mongolian participants were recruited from Barcelona, San Sebastian, and Madrid in Spain; what are the specific numbers of participants from these three cities? Curiously, could there be any difference in infection rates of viral hepatitis between cities?
Barcelona recruited 39 subjects, Madrid recruited 50, and San Sebastian recruited 133 subjects. Thank you for your suggestion, but no infection rate differences has been observed between cities.
- The translated questionnaire is required to be presented as supplementary material for this study.
Thanks for the suggestion. Translated questionnaire has been added as supplementary material.
- Is it possible to conduct any statistical analyses in Table 1? Which would enrich the results section.
Statistical analysis has been performed between those who tested negative and those who tested positive for any serology. P values have been added in Table 1, as well as the Methodology section has been updated. Thanks for the suggestion. Let us know if you believe this enriches de paper.
- Since this study focuses on viral hepatitis, why did the authors not include hepatitis A and E viruses?
The program was focused on hepatitis B, C and D screening in the community. since these infections are the most frequently chronic. Even though hepatitis E can become chronic in immune-supressed subjects, the screening purpose was to diagnose chronically-infected subjects and offer them linkage to care.
Please see the attachment for main text changes.
Many thanks

Reviewer 2 Report
Comments and Suggestions for Authors
General comment
This short communication describes a very laudable effort to elucidate the health burden caused by blood transmissible hepatitis viruses in a cohort of a highly exposed immigrants to a low prevalence country of the European Union. The study design is logical and adequate for the purpose. The number of participants is somewhat low but sufficient for most of the conclusions.
The text is very well written. However, as pointed out below, several minor details need attention and some questions which are left open should be briefly discussed.
Specific points
1. Title. What does D mean in D-Mongolia?
2. Is the limit of detection of the spot test for HBsAg and anti-HCV and of the PSC test for HBV DNA, HCV RNA and HDV RNA known? If so, it should be reported.
3. There is a certain contradiction between the text of the abstract, the discussion and the text of the results section.
a. Abstract (and similarly discussion): Testing found 14 (6.3%) anti-HCV-positive individuals, 2 with detectable HCV RNA (0.9%) and 9 HBsAg-positive (4.1%), 8 with detectable HBV DNA (3.6%). Two individuals had hepatitis B/D co-infection (0.9%).
b. Results section: Among all participants, 13 (5.8%) were anti-HCV positive and 1 (0.5%) had detectable HCV RNA; 8 (3.6%) were HBsAg positive and 7 (3.1%) had detectable HBV DNA. One individual had HBV/HCV co-infection with detectable HBV DNA and HCV RNA. Two of the 9 HBsAg-positive participants tested positive to anti-HDV (25%); HDV RNA was undetectable in both cases.
c. Comments: The data must agree in abstract and results. Furthermore: 2/9 is 22 %, not 25 %.
4. Table1. Should HTA be HT?
5. The methods section mentions ALT, AST, GGT testing. How many participants were tested and how many had elevated levels? Did elevated levels correlate with the markers of active hepatitis?
6. Discussion:” Overall, the prevalence of viral hepatitis found in this study is lower than the rates reported in Mongolia.” Question: Is the difference statistically significant?
7. Discussion: “After the video, there was a 1.6-point improvement in correct answers on the knowledge questionnaire,” What does that mean? 1.6 fold increase in points or scores?
8. Discussion, end: “…a large number of adults were unaware that they had been vaccinated against HBV.” How was the vaccination status inquired if the participants were unaware of their status? Table 1 does not provide data on this. Was an anti-HBs test done?
9. Typo: Abbott, not Abbot;
Author Response
Dear reviewer,
Thank you for your thoughtful response. We are pleased to know that our brief report is being considered for publication in your journal. We have carefully reviewed our manuscript according to your comments, which we believe have contributed to enriching the paper. Below you will find our point-by-point response to your suggestions, as well as the corresponding changes in the full text.
Reviewer #2: This short communication describes a very laudable effort to elucidate the health burden caused by blood transmissible hepatitis viruses in a cohort of a highly exposed immigrants to a low prevalence country of the European Union. The study design is logical and adequate for the purpose. The number of participants is somewhat low but sufficient for most of the conclusions. The text is very well written. However, as pointed out below, several minor details need attention and some questions which are left open should be briefly discussed.
Specific points
- What does D mean in D-Mongolia?
This project obtained a competitive grant from Gilead Sciences, and it was initially intended for hepatitis D screening among Mongolians living in Spain. Therefore, the D in the title stands for hepatitis D. However, the program derived into hepatitis B, C and D screening, so we agree that this is not intuitive and it may lead to confusion. Title has been changed.
- Is the limit of detection of the spot test for HBsAg and anti-HCV and of the PSC test for HBV DNA, HCV RNA and HDV RNA known? If so, it should be reported.
Following the manufacturer reports, limit of detection for the used HBsAg rapid test is 0.1 IU/ml, and antiHCV rapid test has 100% sensitivity and 100% specificity. Limit of detection for PSC’s HBV DNA is 2000 IU/ml, HCV RNA is 1000. This has been added in the Methods section (page 4 lines 90 and 99). Moreover, all subjects with a positive serology underwent a simultaneous regular blood-extraction analysis for serological and virological parameters so appropriate diagnosis could be ensured.
- There is a certain contradiction between the text of the abstract, the discussion and the text of the results section.
- Abstract (and similarly discussion): Testing found 14 (6.3%) anti-HCV-positive individuals, 2 with detectable HCV RNA (0.9%) and 9 HBsAg-positive (4.1%), 8 with detectable HBV DNA (3.6%). Two individuals had hepatitis B/D co-infection (0.9%).
- Results section: Among all participants, 13 (5.8%) were anti-HCV positive and 1 (0.5%) had detectable HCV RNA; 8 (3.6%) were HBsAg positive and 7 (3.1%) had detectable HBV DNA. One individual had HBV/HCV co-infection with detectable HBV DNA and HCV RNA. Two of the 9 HBsAg-positive participants tested positive to anti-HDV (25%); HDV RNA was undetectable in both cases.
- Comments: The data must agree in abstract and results. Furthermore: 2/9 is 22 %, not 25 %.
We agree, it can be easily perceived as a contradiction. One subject had hepatitis B/C co-infection; in the abstract, positive testing determinations were considered, while in the results section, positive patients were considered. It has now been clarified in order to avoid confusion (page 1, abstract)
Moreover, the mistaken percentage you mentioned has been also corrected (page 4).
- Table1. Should HTA be HT?
Acronym has been substituted by Hypertension.
- The methods section mentions ALT, AST, GGT testing. How many participants were tested and how many had elevated levels? Did elevated levels correlate with the markers of active hepatitis?
Transaminase levels were only assessed in those with a positive serology. Elevated AST/ALT levels were only observed in the HBV/HCV co-infected patient with positive HBV DNA and HCV RNA (AST/ALT 37/48 IU/ml).
- Discussion:” Overall, the prevalence of viral hepatitis found in this study is lower than the rates reported in Mongolia.” Question: Is the difference statistically significant?
This is a very interesting question, as it would really validate our results. Unfortunately, we do not have access to the raw data of the referenced study, therefore statistical analysis is not suitable to be performed in this case.
- Discussion: “After the video, there was a 1.6-point improvement in correct answers on the knowledge questionnaire,” What does that mean? 1.6 fold increase in points or scores?
The knowledge questionnaire has 8 questions, each correct answers counting as 1 point. The 1.6-point improvement is the rise in the average correct answers of the knowledge questionnaire. It comes from the subtraction of the mean number of correct answers before (4.7 points), and after (6.3 points) the audiovisual. This has been clarified in the main text (page 4, last paragraph).
- Discussion, end: “…a large number of adults were unaware that they had been vaccinated against HBV.” How was the vaccination status inquired if the participants were unaware of their status? Table 1 does not provide data on this. Was an anti-HBs test done?
Subject’s HBV vaccination status was collected through to the epidemiology survey. No anti-HBs was performed for logistic purposes. A comment has been added at the end of the discussion and in Table 1.
- Typo: Abbott, not Abbot;
Corrected in page 3.
Please see attached file for main text reference.
Many thanks

Round 2
Reviewer 1 Report
Comments and Suggestions for Authors
The authors have adequately addressed my queries in the revised manuscript. I am satisfied with the modifications made by the authors and have no further critical comments.